# Plant Growth Regulators Application Enhance Tolerance to Salinity and Benefit the Halophyte *Plantago coronopus* in Saline Agriculture

**DOI:** 10.3390/plants10091872

**Published:** 2021-09-10

**Authors:** Milagros Bueno, María del Pilar Cordovilla

**Affiliations:** 1Plant Physiology Laboratory, Department Animal Biology, Plant Biology and Ecology, Faculty of Experimental Science, University of Jaén, Paraje Las Lagunillas, E-23071 Jaén, Spain; mpilar@ujaen.es; 2Center for Advances Studies in Olive Grove and Olive Oils, Faculty of Experimental Science, University of Jaén, Paraje Las Lagunillas, E-23071 Jaén, Spain

**Keywords:** antioxidants, climate change, growth, osmolyte accumulation, phytohormones, polyamines, salicylic acid, biosaline agriculture, salt tolerance

## Abstract

Climate change, soil salinisation and desertification, intensive agriculture and the poor quality of irrigation water all create serious problems for the agriculture that supplies the world with food. Halophyte cultivation could constitute an alternative to glycophytic cultures and help resolve these issues. *Plantago coronopus* can be used in biosaline agriculture as it tolerates salt concentrations of 100 mM NaCl. To increase the salt tolerance of this plant, plant growth regulators such as polyamine spermidine, salicylic acid, gibberellins, cytokinins, and auxins were added in a hydroponic culture before the irrigation of NaCl (200 mM). In 45-day-old plants, dry weight, water content, osmolyte (sorbitol), antioxidants (phenols, flavonoids), polyamines (putrescine, spermidine, spermine (free, bound, and conjugated forms)) and ethylene were determined. In non-saline conditions, all plant regulators improved growth while in plants treated with salt, spermidine application was the most effective in improving growth, osmolyte accumulation (43%) and an increase of antioxidants (24%) in *P. coronopus*. The pretreatments that increase the sorbitol content, endogenous amines (bound spermine fraction), phenols and flavonoids may be the most effective in protecting to *P. coronopus* against stress and, therefore, could contribute to improving the tolerance to salinity and increase nutritional quality of *P. coronopus*.

## 1. Introduction

The increase in the world’s population, intensive agriculture, poor quality irrigation water, the decrease in the amount of arable land, desertification, soil salinization, and climate change are all factors that have provoked a decrease in crop quality and yields; therefore, application of innovative techniques could improve crop performance [1,2,3]. Glycophytes are normally used in agriculture, but in a saline environment, they are subjected to osmotic stress and ionic toxicity, factors that negatively affect germination, growth, and crop yield; thus, identifying alternative salt-tolerant crops that can facilitate ecological rehabilitation and restoration and biosaline agriculture should be a priority research area in current agriculture (http://www.sussex.ac.uk/affiliates/halophytes, accessed on 14 June 2021) [4]. Plants halophytes thrive in saline habitats, and can survive in extreme conditions (arid inlands, subtropical habitats, and temperate zones) [5,6]; in addition, these plants possess a series of strategies at anatomical, morphological, physiological, biochemical, and genetic level that allow them to survive to different habitats [7,8]. These strategies are wide-ranging and include phenotypic plasticity, dilution or salt excretion (succulence, salt glands, bladder hairs), decreased transpiration, stomatic and CO_2_ resistance control, water-use efficiency, C3-C4-CAM pathway, high K^+^/Na^+^ compartmentalization (through the Na^+^/H^+^ antiporter of tonoplast and plasma membrane), osmolyte accumulation (polysaccharides, amino acids, polyols), antioxidant systems activation (for protection of photosynthetic apparatus, biomembranes and nucleic acids), the modulation of plant growth regulators, and the expression of certain gene (up-regulating osmolytes and antioxidants) that allows them to survive in a wide variety of environmental conditions [7,8,9,10,11]. On the other hand, halophytes can be used directly as a possible alternative to glycophytes, biofuel-producing crops, fodder and animal feeds, oilseeds and proteins crops, medicinal plants, and in phytoremediation [12,13,14,15,16,17,18,19]. Biosaline agriculture has three main advantages: the recovery of saline and degraded soils, its ability to use wastewater from agriculture, and the increase in the production of metabolites with better nutritional quality [12,13,14,16,17,18,19,20].

In general, plant growth regulators (PGRs) are used to improve crop production and increase to abiotic stress tolerance in glycophytes [21,22]. In saline conditions, PGRs could improve halophytes tolerance for a better crop production. Nevertheless, little is known about PGRs in halophytes and their responses to abiotic stress [23]. These compounds modulate different stages from seed germination to fruit development, ripening and senescence. They also are related to abiotic stress tolerance, and regulate the root: shoot ratio, control stomatal resistance, regulate antioxidant enzymes, delay leaf senescence and act as signal molecules [21]. Auxins regulate cell elongation, vascular tissue development and apical dominance [24]. Cytokinins control cell division, chloroplast biogenesis, leaf senescence, shoot differentiation, anthocyanin production and photomorphogenic development [25]. Gibberellic acid induces seed germination, leaf and stem elongation, favours flowering and fruit development [26,27]. Polyamine application [putrescine (Put), spermidine (Spd), and spermine (Spm)] in agricultural crops serve to protect plants against stress, modulating the homeostasis of reactive oxygen species (ROS), regulating antioxidant systems, cation transport across plant membrane, osmoregulation, and directly or indirectly regulate gene expression [28,29]. Finally, salicylic acid treatment favours the accumulation of osmolytes, alleviates photosynthesis and enhance the upregulation of antioxidant systems in some species [30,31]. We focused our study of PGRs irrigation on the cultivation of *Plantago coronopus*, a halophyte native to the Mediterranean region (South Spain) [32].

*Plantago coronopus* L. (Family Plantaginaceae) inhabits marine cliffs, marshes, and endorheic basins at altitudes up to 800 m (a.s.l.). This halophyte is annual or biennial, with leaves with central veins arranged in basal rosettes measuring 2–20 cm length. Its flowers are produced in spikes and appear in April–October; its seeds are small and brown. It is typically found in saltmarshes in SE Spain [33]. This plant has photosynthesis pathway C3, osmolytes (sorbitol and proline) [34], and antioxidants (phenols and polyamines) [32]. Its mechanisms of tolerance to salinity have been investigated by several authors [32,34,35,36]. Transport of toxic ions (Na^+^ and Cl^−^) to aerial part, and their accumulation in vacuole, in addition to osmotic adjustment in its cytoplasm due to high concentrations of osmolytes allow develop succulence and therefore tolerate a certain degree of salinity [34]. On the other hand, this halophyte is used in biosaline agriculture as its edible leaves are greatly appreciated in salads due to their mild salty taste, crunchy texture, and excellent nutritional value [high content of phenols, amino acids (phenylalanine, tyrosine) and minerals (potassium, calcium, magnesium, sodium, etc.). *Plantago coronopus* showed a higher chlorophyll and flavonoids contents when it was grown in a Se enriched medium. These microgreens showed better nutraceutical value. On the other hand, these herbs grown in the open air presented a better development that in greenhouses, demonstrating the potential of this halophyte in saline agriculture [20,35,37,38,39,40,41].

The following PGRs were added to a hydroponic culture of *P. coronopus*: auxins (indole-acetic acid), cytokinins (Kinetin), gibberellic acid (GA_3_), polyamine (spermidine) and salicylic acid before NaCl (200 mM) application. After 21 days of growth in the absence or presence of salt, dry weight, water content, sorbitol, phenols, flavonoids, endogenous polyamines [putrescine, spermidine, spermine (free, conjugated and bound)], and ethylene were determined. We wanted to identify which PGRs produced the best results to investigate: (1) ways to improve its tolerance to salinity, (2) boost its growth, and (3) increase the nutritional quality of this species. The results could provide technical guidance for increasing the cultivation of this halophyte and the benefits that it provides.

## 2. Results

### 2.1. Effect of Plant Growth Regulators (PGRs) Application on Growth of P. coronopus

Previous works by our research group showed that *P. coronopus* seeds collected from Brujuelo saltmarsh in Jaén (Spain) and cultivated hydroponically showed similar dry weight at 0 and 100 mM NaCl and a decrease at 200 mM NaCl (whole plant) [32]. We decided to choose 0 mM and 200 mM NaCl for the cultivation of this halophyte. Growth parameters such as dry weight and water content, at 45 days old, are shown in Table 1. In non-saline conditions, a positive effect on stem + leaves dry weight (SLDW) and root dry weight (RDW) was observed; being pretreatments Spd and SA (*p* ≤ 0.05) whose having the highest values, above all in RDW (A in Table 1). In the case of Spd, the increases in SLDW and RDW were 47% and 86%, respectively. In water content Spd and SA also had the highest values of all studied pretreatments, especially in roots (increase of 9% compared to control). In saline pretreatments (B in Table 1), Kinetin + salt and Spd + salt obtained the best results for SLDW, while IAA + salt and Spd + salt had the best values for RDW. In the pretreatment Spd + salt the increases were 174% and 197% for SLDW and RDW, respectively, compared to the controls (salt). Growth with the treatments Kinetin + salt and Spd + salt are shown in Figure 1.

### 2.2. Effect of PGRs Application on Sorbitol Content

It is well known that soluble carbohydrates (sorbitol) are plentiful in the family *Plantaginaceae*. For this reason, in the leaves of *P. coronopus* this osmolyte was analyzed at 45 days of culture. In pretreatments without salt (Figure 2A) no significant differences were found between PGR pretreatments compared to the control (without PGRs). However, in saline conditions the Spd + salt and Kinetin + salt had higher values (*p* ≤ 0.05) of osmolyte accumulation (Figure 2B), (increase 0.43-fold and 0.33-fold) respectively, compared to untreated plants (without PGRs + salt). It should also be noted that in all pretreatments under both saline and non-saline conditions, sorbitol concentrations were high even in the control treatments (no PGRs) and (salt).

### 2.3. Effect of PGRs Application on the Total Amount of Phenols and Flavonoids in Saline Conditions

We studied the effect of PGRs application with salt on the antioxidant content (measured as total phenols and flavonoids) in the leaves of *P. coronopus* at 45 days of culture to observe whether any pretreatment PGRs increased phenols and flavonoids content. The results shown in Table 2 indicate that treatments with Kinetin + salt and Spd + salt significantly (*p ≤* 0.01) increased the content of phenols and flavonoids by 24% compared to untreated plants (only salt). The values of phenols and flavonoids under non-saline conditions did not have relevant results or show any significant differences between pretreatments (data not shown).

### 2.4. Effect of PGRs Application on Endogenous Free, Bound and Conjugated Polyamines and Ethylene

In general, the pretreatments Spd without salt, and Spd with salt gave the greatest growth results in *P. coronopus*. Therefore, we considered it necessary to analyze PGRs application on the endogenous PA content (free, bound and conjugated) in the absence or presence of salt. The data are shown in Figure 3, Figure 4 and Figure 5. In salt-free PGRs pretreatment, endogenous Put, Spd, and Spm (free, bound, and conjugated forms) increased compared to the control (-PGRs); the pretreatments with Kinetin, Spd, and SA had the highest values for endogenous Put (Figure 3a), endogenous Spd (Figure 4a) and endogenous Spm (Figure 5a), which corresponded to a greater increase in DW and WC for *P. coronopus*. This increase mainly occurs in bound and free PA fractions. However, under saline conditions, PA levels are modulated by salt. We detected a decreased of endogenous Put (free, bound and conjugated) in pretreatments Kinetin + salt and Spd + salt (Figure 3b) compared to values in Figure 3a; nevertheless, no significant difference was observed in pretreatments with salt due to the low amount of Put detected. However, endogenous Spm did increase in free and, above all, bound forms (Figure 5b), and these values being always higher than observed in saline-free pretreatments. The most significant increase was observed for pretreatment Spd with salt: where endogenous Spm increased two-fold (free form), 2.7-fold (bound form) and 2-fold (conjugated form) compared to the control salt (Figure 5b). Therefore, pretreatments Spd + salt and Kinetin + salt decreased endogenous Put (free, bound and conjugated) and increased endogenous Spm content (above all endogenous Spm bound). In pretreatment Spd + salt the increase of endogenous Spm (bound fraction) (Figure 5b) was higher by 5.3-fold than endogenous Spm (bound fraction) in pretreatment Spd (Figure 5a).

Table 3 shows total PAs (Put (free, conjugated and bound forms) + Spd (free, conjugated and bound forms) + Spm (free, conjugated and bound forms)) under saline and non-saline conditions. All pretreatments increased PA content, especially under saline conditions, the highest values being Spd − salt and Spd + salt. Ethylene production in the leaves of this halophyte were compared to the total PAs. The results indicated a decrease in ethylene production that may contribute to increase PA content due to sharing a common synthesis pathway that we explain in discussion.

The correlation between Spm and sorbitol (r = 0.8465; *p* ≤ 0.01), the total PAs and Spd (r = 0.9193; *p* ≤ 0.01), the total PAs and Spm (r = 0.7184; *p* ≤ 0.01) were always positive. However, the negative correlation between ethylene (C_2_H_2_) and Spm (r = −0.732; *p* ≤ 0.01) and ethylene and total PAs (r = −0.723; *p* ≤ 0.01) indicated that these metabolites (sorbitol. Spd and Spm) are necessary, especially under saline conditions, for enhancing salt tolerance and mitigating the adverse effect of stress (Table 4).

Parameters studied: SLDW (stem + leaf dry weight); RDW (root dry weight); SLWC (stem + leaf water content); RWC (root water content); SOR (sorbitol); PUT (Free + Bound + Conjugated); SPD (Free + Bound + Conjugated); SPM (Free + Bound + Conjugated); Total PAs (Total PUT + Total SPD + Total SPM); C_2_H_2_ (ethylene production).

## 3. Discussion

The benefits that PGRs application have on growth and abiotic stress are well known [42,43,44]. Under saline conditions, PGRs alleviate the adverse effects of salt on morphological, physiological, biochemical characteristics, and on crop yields and quality [10,29,45].

Previous studies showed a fall in dry weight at 200 mM NaCl in *P. coronopus* [32]. This species is in fact less salt-tolerant that other halophytes such as *Frankenia pulverulenta* and *Atriplex prostrata* that also grow in the Brujuelo saltmarsh (Jaén, Spain) [32]. In salt-free pretreatment PGR, the dry weight and water content were increased in *P. coronopus*, especially under pretreatments Spd, SA, and Kinetin in aerial parts and Spd, SA, and IAA in roots (A in Table 1). More specifically, PAs such as Spd are aliphatic biogenic amines. These amines serve as an N reserve for the plant, N:C ratio regulate, favour synthesis of pigments photosynthetic, acid nucleic and proteins. Spermidine applications elevate levels of endogenous PA, but the enzymes involved in its biosynthesis can be increased without altering PA degrading enzymes, such as occurs in zoysia grass subjected to saline stress [46], and therefore these triamine could improve photosynthetic activity and protein synthesis favouring the growth of *P. coronopus* [28,47,48]. On the other hand, SA is a phenolic secondary metabolite, although is more related to abiotic stress tolerance and defensive responses against pathogens, application SA can have beneficious effect on cell and vegetative growth, photosynthesis, and flowering in this halophyte [49]. Regarding auxins and CKs stimulating elongation, cell division, formation of roots, leaves elongation, chloroplast differentiation and photosynthesis, however, a partial effect on growth (IAA stimulated roots and Kinetin stimulated the aerial part) was observed in *P. coronopus*. Crosstalk interaction with other phytohormones, as well as signaling network are very complex. On the other hand, CKs and auxins can have antagonistic effect at low to medium concentration, and only at higher concentration they have adjunctive effect [24,50]. With respect to gibberellins, little effect has in this halophyte, so the effect of each treatment may be genotype-dependent [48]. In saline conditions, the pretreatment Spd with salt was the most effective both in terms of dry weight and water content (B in Table 1). At cellular level, PAs can act as a compatible solute, as scavengers of free radicals, regulate plant membrane transport and act as a signal molecule during stress response [28,51,52,53]. In plant growth, PAs can offer specific protection to the photosynthetic apparatus (structural organization and functional activity of thylakoids), stabilization of biomembranes, and homeostasis redox [54]. A positive effect on photosynthetic activity and uptake of water seems to occur in *P. coronopus* when Spd was applied (Figure 1). Few studies have ever examined PAs in other halophytes. In crops with high nutritional values such as quinoa (*Chenopodium quinoa*), PAs (especially, an increase in Spd and Spm under saline conditions) may be useful markers of salt-tolerant genotypes [55,56] and may exert a protective effect improving growth on *Cymodocea nodosa* [57] and *Solanum chilense* [58]. Specifically, exogenous application of Spd in *C. nodosa* improving chlorophyll fluorescence levels under different saline treatments, maintaining the photosynthetic apparatus functional, under long-term hypo-osmotic stress [57]. Nevertheless, the positive effect of PAs may vary depending on the type of biotic and abiotic stress, plant species, time of exposure and physiological status of the tissues/organs [59,60], and therefore the effect of pretreatments must be studied in each halophyte.

Halophytes (dicotyledonous) accumulate inorganic ions (mainly Na^+^, Cl^−^) in their aerial parts and excrete excess salt through saline glands, bladder hairs or by developing succulence in their leaves [61]. For this reason, we focused our studies on *P. coronopus* leaves. In previous studies, we detected a high concentration of ions (Na^+^, Cl^−^) related to a certain degree of succulence [32]. Al-Hassan et al. [34] concluded that family *Plantaginaceae* have a “constitutive mechanism” of tolerance in which the transport of Na^+^ and Cl^−^ ions (inorganic osmolytes) to the leaves and compartmentalization in the vacuole, contribute to cellular osmotic balance, and increase antioxidant metabolism under saline stress [34,36]. Polyamines are related to ionic transport through at membrane thylakoid, tonoplast, and plasma membranes [52,62,63]. Pottosin and Shabala showed that exogenous PAs application (0.1–1 mM) activated Ca^2+^ efflux, net H^+^ fluxes, and activated H^+^-ATPase pump under stress, but all these experiments were realized in the roots of glycophyte seedlings [52,62,63]. There are no studies on the application of PAs in halophytes on membrane ion channels. Nevertheless, irrigation for 10 days with PGRs (in saline and non-saline conditions) did not modify significantly ionic content and the “pre-adaptation” to stress proposed by Al-Hassan et al. [34] in *P. coronopus* (therefore ion data were not included). On the other hand, the osmoprotective compounds (proline, glycine-betaine, sugar, and polyols) favor water uptake, act as chaperons to molecular stabilized proteins and membranes, scavenge ROS, and/or protect antioxidant enzymes [64,65]. The family Plantaginaceae preferably accumulates sugars and polyols, sorbitol being the most abundant soluble carbohydrate in all *Plantago* species [66]. Sorbitol accumulation and synthesis is carried out above all under anaerobic conditions such as those present in saltmarshes, with confers on the competitive advantages in the environments in which this halophyte normally grows (e.g., saltmarshes in Jaén, Spain) [32,34]. In *P. coronopus*, sorbitol was found in high concentrations in both saline and non-saline PGR pretreatments (Figure 2), although in pretreatments with salt, the Spd had higher values according to higher increase in dry mass and water content. The osmolyte content is probably modulated by PGRs in saline conditions. Pretreatment PGRs stimulate growth probably because they increase photosynthetic activity (above all pretreated with Kinetin and Spd) and increase the sugar content; of these sugars sorbitol plays the role of osmolyte in *Plantago* species growing in adverse environmental conditions. Sorbitol acts to maintain osmotic homeostasis, scavenging ROS, can regulate the osmotic balance, and sequester Na^+^ in the vacuole or apoplast alleviating the toxic effect of saline stress on *P. coronopus* [34]. On the other hand, CKs and PAs mutually regulate different physiological and biochemical processes with strong correlations between CK and PA levels, and act as inter- and intracellular messengers regulating abiotic stress [67].

The selection of productive, fast-growing halophytes with high saline tolerance that give high yields is of vital importance if agriculture is to be successful. *Plantago coronopus* is a source of valuable secondary metabolites of great economic value [37]. Antioxidants such as phenols and flavonoids are an essential part of the human diet and so we used different PGR pretreatments to analyze these two metabolites under saline conditions (Table 2). Previous studies have demonstrated an increase in total phenols as NaCl application increases [32]. These bioactive molecules eliminate large amounts of ROS and protect the cell against oxidative stress on its lipids, proteins, and DNA, in addition act as hydrogen donors, single oxygen quenchers and reducing agents [32,65,68]. The experiments by Boestfleisch et al. [20] have shown that it is possible to manipulate a plant’s antioxidant capacity by modifying the saline growth environment, and the development stage. Our results indicate that mixing Spd with salt significantly improved the content of phenol and flavonoids when compared to untreated plants (only salt). Wild edible plants tend to have higher micronutrient contents and secondary metabolites than those of domestication varieties, therefore *P. coronopus* cultivation irrigated with Spd with salt can increase metabolite contents and constitute a good a source of sugar, minerals, vitamins, and antioxidants, might provide health benefits, and could be used as a new gastronomic food [69,70,71].

The best treatment under both non-saline and saline conditions was the PA Spermidine. Thus, we decided to analyze the endogenous PA content in this halophyte. In the biosynthetic pathway precursors of diamine Put are ornithine and arginine, while the triamine Spd and tetramine Spm are produced by addition of aminopropyl groups from S-adenosyl methionine (SAM) that are sequentially incorporated to Put and Spd by enzymatic reactions catalyzed, respectively, by Spd synthase and Spm synthase. The SAM is decarboxylated by SAMDC (S-adenosyl-methionine decarboxylase) [28,47,53]. Currently, little is known about the endogenous content of PAs in halophytes. Only thirteen halophytes have been studied and PAs have been associated with saline excretion, ionic balance, osmoregulation, protective role on photosynthetic apparatus and biomembranes, high photochemical efficiency in photosystem II, and an increased antioxidant defence system [32,72]. The low levels of free PAs (Put, Spd and Spm) detected under saline conditions in *P. coronopus* [32] made it interesting to study the interconversion between different PA forms under treatment with PGRs. Polyamines can exist in free soluble forms, conjugated to hydroxycinnamic acids (small molecules), or bound to macromolecules such as DNA, lipids, and proteins [51]. In the vegetative stage, salt modulated PA levels, decreased Put content, and increased free and bound forms of Spd and Spm, with values that were always higher than under non-saline conditions. It is interesting underline the drastic increase in bound > free > conjugated Spm forms compared to endogenous Spm (free, bound, and conjugated) in non-saline conditions in pretreatment Spd + salt and pretreatment Spd (Figure 5A,B). We hypothesize that bound forms (above all in Spd pretreatments) can be related to the protection of endogenous cellular structures (mainly biomembranes and photosynthetic apparatus), such as occur in the halophyte *Inula crithmoides* [54]. In addition, Spd treatment increases it endogenous content and enhance also endogenous Spm. More specifically, bound Spd and Spm forms were detected in PSII and LHCII (light-harvesting antenna complex) [73,74]. There are no studies on halophytes, but the exogenous Spd application in some glycophytes showed stabilization of PSII, improving photosynthetic performance and the antioxidant system in chloroplasts under saline conditions [74,75,76]. We consider that similar effects can occur in *P. coronopus* when Spd is applied. At the level of transgenic plants, the Spd synthase gene (*EsSPDS1*) (an enzyme that synthesizes Spd and increase the content of endogenous Spd and Spm) was cloned and characterized in the obligate halophyte *Eutrema salsugineum* and inserted into transgenic tobacco plant subjected to water and salinity stress. The results showed lower malondialdehyde (MDA, oxidative stress indicator) levels, less ion leakage and ROS levels, which indicates better protection in biomembranes, higher water content and more antioxidant enzymes than in non-transformed plants [77]. Clearly, Spd application improves stress tolerance, probably by protecting membranes and photosynthetic apparatus, and decreasing ROS, which could explain our results regarding the enhanced salinity tolerance in *P. coronopus*.

Ethylene and PAs have a common precursor, SAM. The increase of total PAs was accompanied by a decrease in ethylene production under different PGR treatments, which could contribute to PA accumulation (Table 3), thereby indicating a certain competition between PAs and ethylene for SAM, the common precursor. Therefore, SAM can be derivative to the formation of PAs, above all, during salt stress [78,79]. The correlation coefficient between studied parameters (Table 4) confirms our results. Finally, studies in transgenic *Arabidopsis* plants (with overexpression of *SAMDC* and, therefore, with high levels of Spd and Spm) under abiotic stress revealed better growth, maintaining higher photosynthetic activity, higher *Fv*/*Fm* and an increase in the P^I^_ABS_ (Performance Index Based on Absorption). The enhancement in P^I^_ABS_ caused a higher efficiency of quantum yield and specific energy fluxes of PSII, and also higher activities of antioxidant enzymes were found in the transformed plant [80].

## 4. Materials and Methods

### 4.1. Plant Material and Growth Conditions

Seeds of *P. coronopus* were randomly collected in September 2016 from Brujuelo saltmarsh (GPS location: 37°52′46″ N, 3°40′11″ W) (province of Jaén, Spain). Seeds were kept dry at 4 °C before being washed with sterile distilled water and sown in Petri dishes at 25 ± 1 °C and a photoperiod of 16 h [32]. After 10 days, the most uniform seedlings were transferred to 1.5 L pots with vermiculite as a substrate. Four seedlings per pot were sown and hydroponically cultivated using Hoagland nutrient solution 50% pH 6.5 ± 0.1 [81]. Plants were watered every two days with Hoagland nutrient solution. The environmental conditions in the growth chamber were the followings: photosynthetic photon flux density (PPFD) 500 µmol photon m^−^^2^ s^−^^1^, 400–700 nm, provided by Sylvania Inc., Danvers, MA, USA, lamps, photoperiod 16h/8h in a day/night cycle, temperature (day) 25 °C ± 1 °C and (night) 16 °C ± 1 °C, and relative humidity of 55–75%.

### 4.2. Experimental Design and Treatments with PGRs and NaCl

In a growth chamber, plants were acclimated (in hydroponic conditions) for two weeks. Subsequently, these plants were treated for 10 days with different growth regulators applied to the nutrient solution when watered. The growth regulators used were the following: auxin: IAA; cytokinin: kinetin; gibberellins: GA_3_; polyamine: Spd; and salicylic acid: SA.

Six treatments were established with six pots for each treatment PGRs 

No PGRsIAA (1 µM)Kinetin (1 µM)GA_3_ (1 µM)Spd (0.5 mM)SA (0.5 mM)

Six treatments were established with six pots for each treatment PGRs + salt

7.NaCl (200 mM)8.IAA (1 µM) + NaCl (200 mM)9.Kinetin (1 µM) + NaCl (200 mM)10.GA_3_ (1 µM) + NaCl (200 mM)11.Spd (0.5 mM) + NaCl (200 mM)12.SA (0.5 mM) + NaCl (200 mM)

Subsequently, these pots were irrigated with two concentrations of NaCl: 0 mM (treatment 1–6) and 200 mM (treatment 7–12). NaCl levels were selected according to previous experiments realized by us [32]. To avoid osmotic shock, NaCl were increased progressively until the final required concentration was reached [32]; After 21 days in saline or non-saline conditions, plants were harvested for further analysis. Plants were 45 days old when harvested (14 days acclimation in pots, then 10 days of pretreatment with PGRs, and finally 21 days under saline or non-saline conditions). Flowering in this halophyte began approximately at 40–45 days old.

### 4.3. Growth Parameters

The following parameters were determined: fresh weight (FW) (roots, stems, and leaves), dry weight (DW) (leaves + stems and roots), and water content (WC) (leaves + stems and roots). To obtain DW, plants were placed in a forced-air oven at 70v °C for 72–96 h until a constant weight was obtained. This material was used to determine sorbitol, phenols, flavonoids, and endogenous PAs (free, conjugated and bound). Water content was calculated following the formula: WC (%) = (FW − DW/FW) × 100, where SL = stem and leaves, and R = Roots [82]. In fresh material (leaves), the ethylene production was determined.

### 4.4. Sorbitol Quantification

Sorbitol (Sor) was analyzed following Hassan et al. [34] for *P. coronopus* leaves (mature plants). For 10 min, dry leaves (45-day-old) were boiled in milliQ water and subsequently filtered with filters (0.22 µm). Afterwards, all samples (grown in absence and presence of salt) were injecting (20 µL) in a Waters 717 autosampler into a Prontosil 120-3-amino column (4.6 × 125 mm; 3 µm particle size). The conditions of isocratic flux were: (1 mL/min) of 85% acetonitrile for 25 min in each run. Sor integration peaks were obtained in the Waters Empower software and the quantification was realized compared with the standard calibration curve. A Waters 1525 HPLC (high-performance liquid chromatography) coupled with a 2424 evaporative light scattering (ELS) detector (Markham, ON, Canada) were used to determinate Sor content. The source parameters of ELSD were gain 75, data rate 1 point per second, nebulizer heating 60%, drift tube 50 °C, and gas pressure 2.8 Kg/cm^2^. All experiments were conducted at room temperature.

### 4.5. Determination of Total Phenols and Flavonoids

The method of Boestfleisch et al. [20] was followed. Leaves dry were incubated (10 min) in methanol (80%) with continuous shaking. Subsequently, samples obtained in saline conditions were centrifugation for 5 min at 15,000× *g* and the supernatant was collected.

The quantification of total phenols was performed following the protocols by Dudonné et al. [83]. One hundred µL of water was pipetted into small tubes. Then were added: blank (80% methanol), or gallic acid standard (5–250 µg mL^−^^1^) or 10 µL of methanolic extract. The reaction is completed with Folin–Ciocalteu reagent (10 µL). After waiting 8 min sodium carbonate (7%) (100 µL) was added. In the dark and room temperature, tubes were incubated for approximately 90–100 min. Total phenols were calculated using a standard curve. The samples at wavelength of 765 nm were measured in a spectrophotometer VARIAN Cary 4000 UV-VIS (Santa Clara, CA, USA).

The quantification of the total flavonoids was performed following Dewanto et al. [84]. In this case, in each tube was added 150 µL of water. Then we added blank (80% methanol), or catechin hydrate standard (0–400 µg mL^−^^1^), or 25 µL of methanolic extract, and NaNO_3_ (3.75%) (10 µL). After waiting 6 min, the reaction was completed with AlCl_3_ (10%) (15 µL). After 5 min of incubation, NaOH (1 M) (50 µL) was added. Total flavonoids were calculated from a standard curve. The samples and the curve standard at wavelength 510 nm. were measured in a spectrophotometer VARIAN Cary 4000 UVA-VIS (Santa Clara, CA, USA).

### 4.6. Analysis of Free, Bound and Conjugated Polyamines

For PAs extraction the method followed by Ghabriche et al. [54] was used with minor modifications. Dry leaf samples (in saline and non-saline conditions) were ground in a mortar and homogenized with HCl (1 M) (*v*/*v*), then centrifuged at 23,000× *g* at 4 °C for 20 min. The supernatant was used to determine free polyamines by dansylation method [85]. The samples were resuspended in methanol (1 mL) and then centrifuged at 13,000× *g* for 15 min. Later, these samples needed to be filtered using microfilters (Chromafil PES-45/15, 0.45 µm; Macherey-Nagel). Twenty µL were injected into a Bio-Rad HPLC system (Hercules, CA, USA) equipped with a Nucleosil 100-5 C18MN 250/04 column (particle size: 5 µm, 4.6 × 250 mm^2^). The conditions of HPLC to quantify the integration peaks were the following: a methanol/water stepped gradient program changing from 60% to 100% methanol over 25 min, flow rate 1 mL min^−^^1^, and temperature of 35 °C. A Shimadzu RF-10Axl fluorimeter detector (excitation wavelength 320 nm and emission wavelength 510 nm) was used to determine dansylated free polyamines.

Bound forms (covalently bound to macromolecules such as proteins) and conjugated forms (covalently bound with small molecules such as hydroxycinnamic acids) were also analyzed. We added 200 µL of HCL (12 N) to the same amount of supernatant (200 µL) and transferred to dark tightly capped glass tubes. These tubes were placed in a heater and heated at 110 °C for 24 h to realize sample hydrolysis. After HCl was evaporated, the residue was resuspended in 200 µL of perchloric acid (10%) and used for dansylation. The pellet was used to extract bound PAs. This was dissolved in 5 mL of NaOH (1N). The mixture was centrifuged at 23,000× *g* at 4 °C for 20 min., and the supernatant was hydrolysed and dansylated under the same conditions described above. Dansylated free PAs (supernatant), dansylated conjugated PAs (supernatant hydrolysed) and dansylated bound PAs (pellet hydrolysed) were injected (20 µL) in the HPLC, in addition to PA standards (Put, Spd, Spm from Sigma, San Francisco, CA, USA), for quantification.

### 4.7. Ethylene Production

The method of Bueno et al. [86] was followed: fresh leaves collected of *P. coronopus* (45-day-old) were immediately transferred into a 5 mL flask (containing at the bottom filter paper and 50 µL of distilled water). All flasks were sealed with a silicone-rubber stopper (to prevent gas leakage). Flasks were incubated on a stove for 1 h incubation period, at 30 °C in darkness. Later, a 1 mL gas sample was injected into a HP 5890 series II, Hewlett Packard (Palo Alto, CA, USA) gas chromatograph fitted with a flame ionization detector and a 2 m × 4 mm stainless-steel column packed with 50–80 mesh Poropack-R. The conditions of chromatograph were: N_2_, H_2_ and synthetic air flow rates 50, 86, and 400 mL min^−^^1^, respectively. To analyze and quantify ethylene production, peaks integration was compared with the retention time of ethylene (C_2_H_4_) standard, (purity 99.9%).

### 4.8. Data Analysis

A randomized block design was used in our experiments. Data are presented as mean ± standard error (SE). A Statgraphics Centurion v. 17 (University of Jaén) was used to perform analyses of variance (ANOVA). Significant differences between means were determined using Tukey’s multiple range test (*p ≤* 0.05 and *p* ≤ 0.01). All parameters in the absence or presence of salt were compared using Pearson’s correlation coefficients.

## 5. Conclusions

Halophyte cultivation as a part of biosaline agriculture could help improve productivity and crop quality and be used to restore saline and degraded land. In *P. coronopus* cultivation, exogenous Spd application (0.5 mM) to the nutritive solution can improve growth and increase salt-stress tolerance, as well as increasing the osmolyte (sorbitol) and antioxidant compounds (phenols and flavonoids) under saline conditions. The increase in the endogenous PA pool, especially Spd and Spm (bound forms), is probably related to the protection of subcellular structures, the maintenance of photosynthetic activity, osmotic adjustment, ionic homeostasis and the improvement of antioxidant activity. In addition, the increase in Spd levels showed a negative correlation with ethylene, indicating than the decrease in ethylene also can contribute to PA accumulation. Auxins, CKs, GAs and SA pretreatments stimulated growth under non-saline conditions, but these PGRs were unable to mitigate the adverse effects of stress. Therefore, Spd application is the best pretreatment for *P. coronopus* cultivation and can contribute to improving the tolerance to salinity and nutritional quality of this halophyte, although it will be necessary to research in each halophyte which is the better treatment to apply.

## Figures and Tables

**Figure 1 plants-10-01872-f001:**
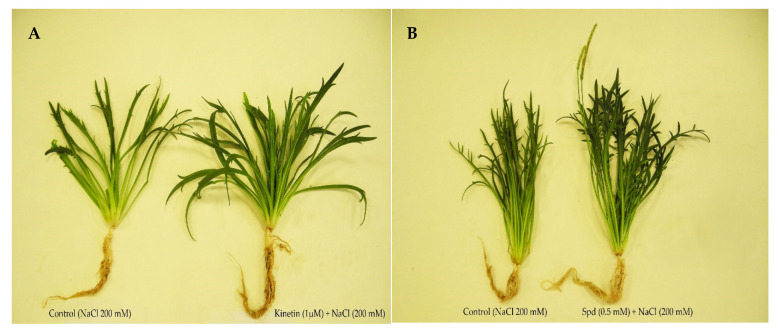
*Plantago coronopus* cultivated with Kinetin + salt compared to control (**A**), and *P. coronopus* cultivated with Spd + salt compared to control (**B**).

**Figure 2 plants-10-01872-f002:**
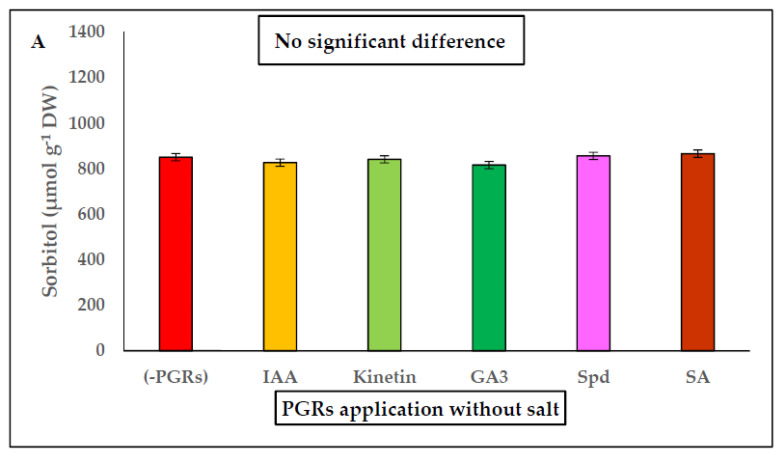
(**A**) Sorbitol content in *P. coronopus* leaves with the following PGRs: IAA, Kinetin, GA_3_, Spd, and SA compared to control (-PGRs). (**B**) Sorbitol content with PGRs in saline conditions: IAA + salt, Kinetin + salt, GA_3_ + salt, Spd + salt, and SA + salt, compared with salt (200 mM NaCl). Means ± SE (*n* = 3). Different letters above bars represent significant difference between treatments (*p* ≤ 0.05).

**Figure 3 plants-10-01872-f003:**
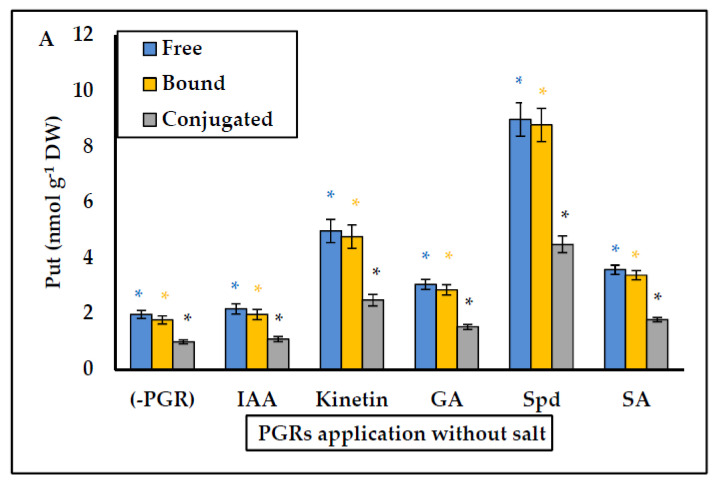
(**A**) Effect of PGRs application without salt, and (**B**) PGRs with salt (200 mM NaCl), in *Plantago coronopus* leaves at 45-day-old, on Putrescine (Free, Bound and Conjugated) content. Means ± SE (*n* = 3). The asterisk above the column represents significant difference between treatments of free Put, significant difference between treatments of bound Put and significant difference between treatments of conjugated Put, according to Tukey’s test (*p* ≤ 0.01).

**Figure 4 plants-10-01872-f004:**
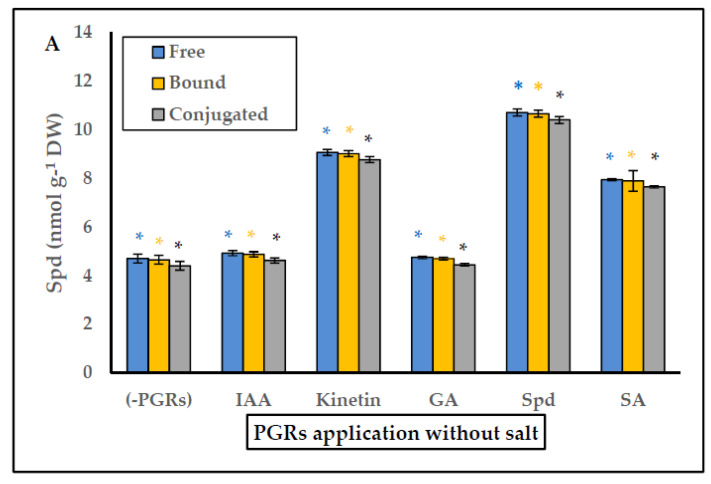
(**A**) Effect of PGRs application without salt, and (**B**) PGRs with salt (200 mM NaCl), in *Plantago coronopus* leaves at 45-day-old, on Spermidine (Free, Bound and Conjugated) content. Means ± SE (*n* = 3). The asterisk above the column represents significant difference between treatment of free Spd, significant difference between treatments of bound Spd and significant difference between treatments of conjugated Spd, according to Tukey’s test (*p* ≤ 0.01).

**Figure 5 plants-10-01872-f005:**
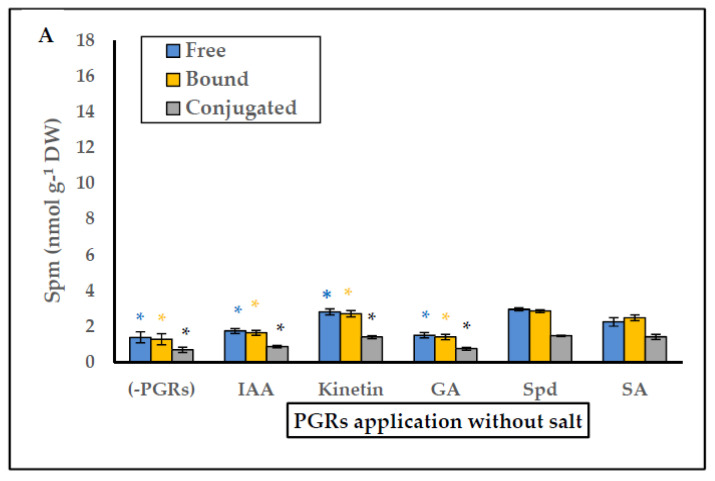
(**A**) Effect of PGRs application without salt, and (**B**) PGRs with salt (200 mM NaCl), in *Plantago coronopus* leaves at 45-day-old, on Spermine (Free, Bound and Conjugated) content. Means ± SE (*n* = 3). The asterisk above the column represents significant difference between treatments of free Spm, significant difference between treatments of bound Spm and significant difference between treatments of conjugated Spm, according to Tukey’s test (*p* ≤ 0.01).

**Table 1 plants-10-01872-t001:** (**A**) Effect of PGRs (plant growth regulators) application in salt-free pretreatment, (**B**) effect of PGRs application under saline conditions (200 mM NaCl) in *Plantago coronopus*, at 45 days of culture on SLDW (stem + leaves dry weight), RDW (root dry weight), SLWC (stem + leaves water content), and roots water content (RWC). Means ± SE (*n* = 16). Different letters within the same row represent significant differences between treatments, according to Tukey’s test (*p* ≤ 0.05).

**A.** **PGRs Application without Salt**	**SLDW** **(g/plant)**	**RDW** **(g/plant)**	**SLWC (%)**	**RWC (%)**
Control (no PGR)	0.134 ± 0.0038 ^c^	0.0198 ± 0.0014 ^b^	94.16 ± 0.47 ^a^	85.07 ± 0.24 ^c^
IAA	0.171 ± 0.0056 ^ab^	0.0321 ± 0.0037 ^a^	95.45 ± 0.52 ^a^	92.05 ± 0.28 ^ab^
Kinetin	0.179 ± 0.0062 ^a^	0.0214 ± 0.0016 ^b^	95.67 ± 0.55 ^a^	90.63 ± 0.63 ^ab^
GA_3_	0.147 ± 0.0052 ^bc^	0.0199 ± 0.0018 ^b^	95.51 ± 0.46 ^a^	89.80 ± 0.47 ^b^
Spd	0.197 ± 0.0089 ^a^	0.0369 ± 0.0021 ^a^	95.91 ± 0.56 ^a^	92.88 ± 0.66 ^a^
SA	0.186 ± 0.0063 ^a^	0.0341 ± 0.0012 ^a^	95.74 ± 0.39 ^a^	92.53 ± 0.45 ^a^
**B.** **PGRs Application with Salt**	**SLDW** **(g/plant)**	**RDW** **(g/plant)**	**SLWC (%)**	**RWC (%)**
Control (salt)	0.080 ± 0.0110 ^d^	0.0101 ± 0.0005 ^d^	91.54 ± 0.35 ^b^	82.63 ± 0.33 ^c^
IAA + salt	0.132 ± 0.0078 ^bc^	0.0167 ± 0.0005 ^b^	92.50 ± 0.38 ^b^	85.54 ± 0.38 ^ab^
Kinetin + salt	0.161 ± 0.0064 ^b^	0.0128 ± 0.0004 ^c^	93.18 ± 0.42 ^b^	83.55 ± 0.90 ^bc^
GA_3_ + salt	0.085 ± 0.0063 ^d^	0.0117 ± 0.0006 ^cd^	92.11 ± 0.52 ^b^	83.09 ± 0.56 ^bc^
Spd + salt	0.219 ± 0.0100 ^a^	0.0300 ± 0.0058 ^a^	95.72 ± 0.45 ^a^	87.55 ± 0.40 ^a^
SA + salt	0.092 ± 0.0090 ^cd^	0.0094 ± 0.0008 ^d^	92.50 ± 0.44 ^b^	82.43 ± 0.70 ^c^

**Table 2 plants-10-01872-t002:** Effect of PGRs application with salt (200 mM NaCl) in *P. coronopus* leaves, at 45 days old, on total phenols and flavonoids. The values represent means ± SE (*n* = 3). The total phenols was expressed in mg gallic acid (GAE) per gr dry weight, and total flavonoids was expressed in mg of catechin (CE) per gr dry weight. Different letters within the same row represent significant difference among treatments, according to Tukey’s test (*p* ≤ 0.01).

PGRs Application with Salt	Total Phenols (mg GAE g^−1^ DW)	Total Flavonoids (mg CE g^−1^ DW)
Control (Salt)	4.5 ± 0.11 ^c^	3.1 ± 0.11 ^b^
IAA + salt	4.9 ± 0.12 ^bc^	3.5 ± 0.12 ^ab^
Kinetin + salt	5.3 ± 0.10 ^ab^	3.7 ± 0.23 ^ab^
GA_3_ + salt	4.6 ± 0.17 ^bc^	2.9 ± 0.23 ^b^
Spd + salt	5.9 ± 0.21 ^a^	4.0 ± 0.14 ^a^
SA + salt	5.0 ± 0.20 ^bc^	3.2 ± 0.20 ^b^

**Table 3 plants-10-01872-t003:** (**A**) Effect of PGRs application without salt, and (**B**) PGRs with salt (200 mM NaCl) in *P. coronopus* leaves on total PAs [Put (Free, Bound and Conjugated) + Spd (Free, Bound and Conjugated) + Spm (Free, Bound and Conjugated)] and ethylene. Means ± SE (*n* = 3). Different letters within the same row represent significant differences among treatments, according to Tukey’s test (*p* ≤ 0.01).

**A.** **PGRs Application without Salt**	**Total PAs** **(nmol g^−1^ DW)**	**Ethylene** **(nL g^−1^ FW h^−1^)**
Control (no PGRs)	21.87 ± 0.68 ^d^	10.30 ± 0.64 ^a^
IAA	24.54 ± 0.75 ^d^	8.35 ± 0.61 ^ab^
Kinetin	46.00 ± 1.81 ^b^	5.62 ± 0.71 ^b^
GA_3_	25.01 ± 0.88 ^d^	7.68 ± 0.67 ^ab^
Spd	61.29 ± 1.45 ^a^	6.35 ± 0.53 ^b^
SA	36.50 ± 0.69 ^c^	5.81 ± 0.65 ^b^
**B.** **PGRs Application with Salt**	**Total PAs** **(nmol g^−1^ DW)**	**Ethylene** **(nL g^−1^ FW h^−1^)**
Control (salt)	32.11 ± 0.58 ^c^	7.61 ± 0.52 ^a^
IAA + salt	49.94 ± 0.42 ^b^	3.30 ± 0.68 ^b^
Kinetin + salt	54.70 ± 2.42 ^b^	3.77 ± 0.47 ^b^
GA_3_ + salt	36.76 ± 1.16 ^c^	5.23 ± 0.59 ^ab^
Spd + salt	67.79 ± 2.54 ^a^	3.74 ± 0.64 ^b^
SA + salt	37.50 ± 1.94 ^c^	5.76 ± 0.53 ^ab^

**Table 4 plants-10-01872-t004:** Simple correlation coefficient (Pearson method) among all parameters studied in saline and non-saline conditions in *P. coronopus* (*p* ≤ 0.05 *; *p* ≤ 0.01 **).

	SLDW	RDW	SLWC	RWC	SOR	PUT	SPD	SPM	Total PAs	C_2_H_2_
SLDW	1									
RDW	0.7522 **	1								
SLWC	0.7629 **	0.7559 **	1							
RWC	0.7049 **	0.8769 **	0.7941 **	1						
SOR	0.0723	−0.2711	−0.2874	−0.440 **	1					
PUT	0.4624 **	0.4943 **	0.3212	0.4720 **	−0.0267	1				
SPD	0.6824 **	0.4669 **	0.3550 *	0.3892 *	0.3662 *	0.7729 **	1			
SPM	0.1200 *	−0.3134	−0.3501 *	−0.470 **	0.8465 **	−0.0308	0.4360 **	1		
Total PAs	0.5148 **	0.2064	0.0816 *	0.0840	0.5897 **	0.6838 **	0.9193 **	0.7184 **	1	
C_2_H_2_	−0.2150	0.1044	0.2573	0.1496	−0.659 **	-0.2616	−0.588 **	−0.732 **	−0.723 **	1

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
