# Peer review of "Plant Growth Regulators Application Enhance Tolerance to Salinity and Benefit the Halophyte Plantago coronopus in Saline Agriculture"

_plants, 2021, doi:10.3390/plants10091872_

Round 1

Reviewer 1 Report

The manuscript titled “Polyamines Application Enhance Tolerance to Salinity and Benefit the Halophyte Plantago coronopus in Saline Agriculture” Manuscript ID: plants-1351640, by Bueno and Cordovilla, provides insight into the potential role of different plant growth regulators (auxins, cytokinins, gibberellins, and salicylic acid) and spermidine, NOT POLYAMINS in general, in mediating the salinity tolerance of the halophyte Plantago coronopus. Although the idea is interesting, it leaks originality/novelty, since the authors just published (less than a month ago) another article in Agronomy (kindly check this link https://doi.org/10.3390/agronomy11081515) that present the same concept, hypothesis, and findings, but on a different halophyte (Frankenia pulverulenta). In my point of view, the current study does not add much value to the field. Moreover, the current manuscript is similar (more than 35%) with the article mentioned above and published by the same authors.

I strongly recommend authors rewrite the MS and illustrate the hypothesis and content clearly in the MS. Moreover, I highly recommend they go deeper to better understand the physiological, biochemical, and molecular mechanisms behind the potential role of these compounds in salinity tolerance. Finally, Although the language used in the manuscript is easy to follow, however, the manuscript should be carefully and deeply revised for grammar and English use, since several mistakes were found throughout the whole paper.

Author Response

Please see the attachment of pdf with responses of the reviewer 1

Reviewer 2 Report

In this article, authors applied plant growth regulators to haplophyte (P.coronopus) and investigated their role in salt tolerance.

My specific comments are;

  • Introduction is very long and can be shortened.
  • Figure 2: Authors should present more data on the morphology of plants after PGRs treatment as it appears that it significantly affecting plant morphology which leads to increase in dry weight.
  • Figures 3: Authors should merge both graphs. In the present form, it is not easy to compare PGRs affect. Whether sorbitoal levels in PGRs-salt (top graph) are statistically different?
  • Table 2: Whether only the application of PGRs affect phenols and flavonoids levels? And why authors did not present the data of only PGRs-salt?
  • Figure 4-6. It is very hard to understand. Perhaps authors should split in to three graphs, i.e, free, bound and conjugated but keep PGRs-salt and PGRs+salt in the same graph.
  • Statistical test (Figure 4-6): it is not clear how authors performed these tests? Did they consider PGRs-salt and PGRs+salt and compare across treatments? It seems that they are just deal them separately.
  • Discussion must be improved and should be more specific and concise.
  • Authors should check the Na+ levels in the root, shoot and leaf. This PGRs affect is due to vacuole sequestration or simply PGRs block the Na+ transport channels. This should be clarified with experiments.

Author Response

We attached a pdf with the responses of reviewer 2

Reviewer 3 Report

  1. Since authors used several plant growth regulators, it is better to use term  plant growth regulator in the title. Remove polyammines as it is misleading the content of manuscript.
  2. Abstract:Add some important data in abstract, in form of %change and add take home message at the end.
  3. Introduction: Too lengthy, reduce it. Also I noticed that flow is lost in the middle of introduction. And I suggest to Improve the hypothesis part. Remove figure 1. 
  4. Tables: letters denoting statistical difference should be in superscript.
  5. Fig 3a, B, better to write "without salt"  "with salt"
  6. Figures, it should be "no significant difference"
  7. Discussion is weak in my opinion. I would suggest authors to discuss molecular mechanisms behind the current results. Also, refer latest literature
  8. Update it with recent literature.
  9. Screen for typo errors, there are many.
  10. conclusion section is well written, but is too long. Reduce it.

Author Response

We attached a pdf with the responses of reviewer 3

Round 2

Reviewer 1 Report

Thanks for providing this information about both halophyte species. However, your response did NOT satisfy me and I still see NO adding value in work in its current form. Moreover, the revised version of the manuscript still has more than 30% similarity/plagiarism (see the attached file), which means that you did NOT even consider my main concern in my previous report.

Author Response

  • Thanks, We have revised the manuscript.

Reviewer 2 Report

Authors have answered all the queries and improved the manuscript.

Author Response

Thanks.

Reviewer 3 Report

Authors have addressed all of my comments.

Author Response

Thanks